# Application of High-Pressure Processing (or High Hydrostatic Pressure) for the Inactivation of Human Norovirus in Korean Traditionally Preserved Raw Crab

**DOI:** 10.3390/v15071599

**Published:** 2023-07-21

**Authors:** Pantu Kumar Roy, Eun Bi Jeon, Ji Yoon Kim, Shin Young Park

**Affiliations:** 1Institute of Marine Industry, Department of Seafood Science and Technology, Gyeongsang National University, Tongyeong 53064, Republic of Korea; vetpantu88@gmail.com (P.K.R.); eunb61@naver.com (E.B.J.); 2West Sea Fisheries Research Institute, National Institute of Fisheries Science, Incheon 22383, Republic of Korea; juniee72@naver.com

**Keywords:** human norovirus GII.4, propidium monoazide, sodium lauroyl sarcosinate, raw crab, high-pressure processing

## Abstract

Human norovirus (HuNoV) is a common cause of outbreaks linked to food. In this study, the effectiveness of a non-thermal method known as high-pressure processing (HPP) on the viable reduction of an HuNoV GII.4 strain on raw crabs was evaluated at three different pressures (200, 400, and 600 MPa). HuNoV viability in raw crabs was investigated by using propidium monoazide/sarkosyl (PMA) as a nucleic acid intercalating dye prior to performing a real-time reverse transcription-polymerase chain reaction (RT-qPCR). The effect of the HPP exposure on pH, sensory, and Hunter colors were also assessed. HuNoV was reduced in raw crabs compared with control to HPP (0.15–1.91 log) in non-PMA and (0.67–2.23 log) in PMA. HuNoV genomic titer reduction was <2 log copy number/µL) when HPP was treated for 5 min without PMA pretreatment, but it was reduced to >2 log copy number/µL after PMA. The pH and Hunter colors of the untreated and HPP-treated raw crabs were significantly different (*p* < 0.05), but sensory attributes were not significant. The findings indicate that PMA/RT-qPCR could be used to detect HuNoV infectivity without altering the quality of raw crabs after a 5 min treatment with HPP. Therefore, HuNoV GII.4 could be reduced up to 2.23 log in food at a commercially acceptable pressure duration of 600 MPa for 5 min.

## 1. Introduction

Human norovirus (HuNoV) is a single-stranded, non-enveloped RNA virus that belongs to the *Caliciviridae* family [1]. Diarrhea, vomiting, nausea, and stomach pain are the most common symptoms of HuNoV infection. HuNoV affects people of all ages, and the gastroenteritis caused by the virus recovers on its own in 1 to 3 days. Interpersonal contact, airborne pathways, and the intake of infected foods are all linked to HuNoV outbreaks, with the GII.4 genotype accounting for 70–80% of all outbreaks [2]. Foodborne transmission of HuNoV includes the ingestion of contaminated seafood from feces in the manufacturing area, but also contaminated fruit and vegetables, bivalve mollusks, and so on [1,3]. HuNoV is a leading cause of non-bacterial acute gastroenteritis, accounting for more than 21 million disease cases each year. Furthermore, HuNoV is a major cause of foodborne illness in the USA, causing approximately 5 million cases each year [4]. Foods that are at a higher risk of contamination by HuNoV include seafood, fresh produce, and ready-to-eat foods. There are several factors that contribute to the susceptibility of seafood to HuNoV contamination. One primary reason is that seafood, particularly shellfish, is often consumed raw or undercooked, which means there is minimal heat treatment to inactivate any viruses present. HuNoV can survive at low temperatures, making it more resilient to typical refrigeration or freezing conditions. Another factor is the natural habitat of certain seafood, such as oysters, clams, and mussels. These filter-feeding shellfish have the ability to accumulate viruses and other pathogens present in the surrounding water, including HuNoV. If the water is contaminated with the virus, the shellfish can become infected, posing a risk to consumers who consume them without proper cooking [1]. Contamination can also occur during the handling and processing of seafood. Improper hygiene practices, such as inadequate handwashing or cross-contamination with surfaces or utensils that have come into contact with the virus, can introduce HuNoV to seafood [5]. In terms of inactivation, foodborne viruses located internally in edible product structures present a unique challenge. To inactivate foodborne viruses while keeping the seafood fresh, an intervention that can target viruses present inside the food matrix must be used.

HuNoV has been successfully detected using real-time quantitative PCR (RT-qPCR) on potentially contaminated food [5]. Therefore, we are trying to identify intact and non-intact virus particles by applying RT-qPCR. Because only the infectious virus can cause foodborne infection, reliable identification of the infectious virus in foods is crucial and necessary to assess the risk of foodborne outbreaks [6]. The current standard technology for detecting HuNoV in foods is RNA-based detection methods. The inability to establish whether the RNA identified is from an infectious or non-infectious viral particle is a significant disadvantage of reverse transcriptase PCR (RT-qPCR)-based technologies. Therefore, the outcomes of these approaches may not accurately indicate the risk of developing HuNoV illness after ingestion. Methods for assessing HuNoV capsid integrity while also detecting genomic RNA have recently been developed. The propidium monoazide (PMA) technique is one such approach [6].

In previous studies, propidium monoazide (PMA) was used to distinguish viable from nonviable virus using the RNA genome in a virus RT-qPCR experiment [1,6]. Additionally, Parshionikar et al. [6] employed a PMA pretreatment methodology to assess the infected and non-infected virus. In food and the environment, PMA pretreatment along with RT-qPCR was found to be successful in recognizing selectively infected murine norovirus (MNV), rotavirus, and hepatitis A virus (HAV), including HuNoV [1,7,8,9]. Therefore, PMA is an appropriate method for predicting HuNoV viability. Under visible light, the monoazide agents can penetrate damaged viral capsids and, subsequently, covalently attach to viral nucleic acids, inhibiting RT-qPCR. As a result, PMA-RT-qPCR has been used to assess the efficacy of chemical or physical therapies for microbial inactivation. The PMA-RT-qPCR assay in particular proved successful in assessing the viability of non-culturable viruses [10]. According to a previous report [11], the Ct values obtained from PMA-RT-qPCR were substantially greater than those obtained with RT-qPCR following heat treatment [10]. Furthermore, detergents such as sodium deoxycholate and sodium lauroyl sarcosinate (INCI) were used; these anionic detergents aid PMA in penetrating injured cells and improving the distinction between dead and live cells [12]. Surfactants can help monoazide penetrate slightly damaged capsids of the hepatitis A virus [13].

High-pressure processing (HPP) is a non-thermal food processing technology widely used in the food industry to extend food’s shelf life and safety [14]. During HPP treatment, pressure is equally dispersed throughout the food product. This allows viruses found within the food matrix as well as on the food’s surface to be inactivated. HPP has been found to be effective in controlling HuNoV in various foods. HPP, for example, has been demonstrated to be effective in controlling HuNoV outbreaks in oysters, a significant high-risk food for HuNoV epidemics. HuNoV GI.1-seeded oysters treated with HPP (600 MPa at 6 °C for 5 min) did not cause infection in human subjects who ate the oysters [15]. The performance of HPP is significantly influenced by the characteristics of the sample matrix. Factors such as pH, salt concentration, fat composition, and protein composition can all have a notable impact on the effectiveness of pathogen inactivation. This observation holds true for viral inactivation as well. Changes in these parameters within the sample matrix can affect the susceptibility of pathogens, including viruses, to HPP treatment [16]. Therefore, it is crucial to consider and account for the specific composition and properties of the sample matrix when determining HPP parameters for effective pathogen and viral inactivation.

In Korea, entire raw, fresh crabs (*Portunus trituberculatus*) with the shell, intestine, and roe are marinated in soy sauce with additional ingredients (such as sesame oil, minced garlic and ginger, and finely sliced scallions), then quickly boiled to kill any lingering germs for short-term preservation of the crabs. Ganjanggejang is a type of Jeotgal, a Korean dish that consists of salted, preserved raw seafood (ganjang, meaning soy sauce; ge, meaning raw crab; and jang, meaning condiments). The maximum shelf life for commercially made ganjanggejang is two weeks when kept in a refrigerator.

Therefore, the objective of this study was to explore the efficacy of HPP treatment against HuNoV GII.4 infectivity in raw crabs using PMA + sarkosyl pretreatment and RT-qPCR. This study also investigated whether the HPP treatment length time (200–600 MPa) influences the pH value, sensory attributes, and Hunter color properties of raw crabs.

## 2. Materials and Methods

### 2.1. HuNoV Stock Preparation

The human norovirus (HuNoV) GII. 4 strain applied in this study was obtained in 2019 at the Gyeonggi Institute of Health and Environment (GIHE; Gyeonggi-do, Republic of Korea) from patients with gastrointestinal symptoms caused by HuNoV [9,11,17]. HuNoV genotype was confirmed and stored in the Waterborne Virus Bank (WAVA; Seoul, Republic of Korea). The HuNoV GII. 4 was obtained from WAVA and was made from a stock of 500 μL of phosphate-buffered saline (PBS; pH 7.2) [17]. Virus strains were transported frozen and packaged in dry ice, then kept at –80 °C for further use in research.

### 2.2. Preparation of Sample and Virus Inoculation

In this experiment, raw crabs (from Tongyeong, Republic of Korea) were obtained from an online seller and shipped frozen. They were immediately transported to the fridge (4 °C) and used in tests within 24 h after delivery. The raw crabs were prepared by extracting and assembling their internal organs, then homogenizing them using a homogenizer (homogenizer stirrer, Daihan Scientific Co., Wonju, Republic of Korea) and dividing them into 3 g samples [9,17]. Every sample was placed in a Petri dish (30 × 30, L × W mm). Each sample was inoculated with 30 μL of HuNoV GII. 4 (6.5 log copy/μL) and (CHC Lab Co. Ltd., Daejeon, Republic of Korea) for 1 h to allow the HNV GII. 4 to absorb the sample in clean bench (CHC Lab Co. Ltd., Daejeon, Republic of Korea) [11,17].

### 2.3. HPP Treatment of HuNoV in Raw Crabs

HPP treatment was used in this experiment on raw crab. The device was turned on at least 10 min before the start of the experiment and the surface of the raw crab samples inoculated with HuNoV GII.4 was treated with HPP for 200, 400, and 600 MPa for 5 min at 6 °C (Petri dish size 35 × 15 mm). Following treatment, the levels of HuNoV RNA were determined using the PMA + sarkosyl binding assay followed by RT-qPCR.

### 2.4. Propidium Monoazide and Sodium Lauroyl Sarcosinate Treatment

To allow for dye penetration, the virus-treated samples were immediately combined with 200 µM PMA (Biotium, Hayward, CA, USA) and incubated in the dark for 5 min at room temperature. Then, samples were photoactivated for 10 min on each side with 40 W LED light (Dynebio, Seongnam, Republic of Korea) at 460 nm wavelength at room temperature and distance was 3 mm from the samples [1]. A control group was included in the study that was not treated with PMA or exposed to halogen light to examine if the dye treatment interfered with viral identification. Sodium lauroyl sarcosinate (Sigma-Aldrich, St. Louis, MO, USA) was used at 1.0% (*w*/*v*) to optimize the sarkosyl concentration and to examine its effect on HuNoV [18]. Sarkosyl solution was added to the PMA mixture at the same time as detergent treatment. PMA + sarkosyl-treated samples were incubated for 10 min at room temperature in the dark. An untreated control was included and unexposed to LED light to examine the ability of PMA + sarkosyl treatment to interfere with HuNoV detection [9,17,18]. Finally, before RT-qPCR experiments for discriminative detection of possibly infectious and noninfectious HuNoV viral particles, viral samples were exposed to the optimized PMA and sarkosyl pretreatments.

### 2.5. Virus Isolation and RNA Extraction

Samples of raw crab homogenates sufficiently infected with HuNoV GII.4 were treated with the proteinase K (PK) method of Jeon et al. [9] In total, 3 g of raw crab homogenates were added to the same volume of PK solution (100 g/mL; Sigma-Aldrich, Dorset, UK) at a final concentration of 0.1 mg/mL. The sample was then shaken for 1 h at 37 °C at 290 rpm. After that, the sample was incubated for 15 min at 60 °C before being centrifuged for 10 min at 10,000× *g* (5400 rpm). The supernatant (approximately 0.4 mL) was collected in a sterile 15 mL tube and stored at –80 °C until RNA extraction was performed. A RNeasy Mini Kit (Qiagen, Hilden, Germany) was used to extract and purify viral RNA in a final volume of 60 μL, as directed by the manufacturer. Following the extraction, HuNoV GII.4 was detected and quantified using RT-qPCR analysis of the sample’s RNA [9,17].

### 2.6. Analysis of HuNoV Using RT-qPCR

A previous study reported [18] reverse transcription was used for cDNA synthesis. For HuNoV GII.4 gene amplification, 1 μL of enzyme mix (reverse transcriptase) (5 units/μL), 1 μL of 10 mM dNTP, 0.25 μL of RNase inhibitor (5 units/μL), 5 μL of 5X RT–PCR buffer (RNeasy Mini Kit; Qiagen, Hilden, Germany), 1 μL of 10 M primer (Forward and Reverse), 5 μL of extracted RNA, and RNase-free water were used to amplify the HuNoV GII.4 gene [1]. This solution had a total volume of 25 μL, which included 5 μL of RNA isolated using the RNA extraction method [17]. The following RT-qPCR amplification protocol was used with a TaKaRa TP800–Thermal Cycler Dice Real-Time System (TaKaRa, Seoul, Republic of Korea): initial denaturation at 95 °C for 10 min, followed by 45 cycles of amplification at 53 °C for 25 s and at 62 °C for 70 s [18]. To boost sensitivity and specificity, primers and probes were constructed for the ORF–1 and ORF–2 overlapping sections. COG2F: 5′–CAR GAR BCN ATG TGG AGR ATG AG–3′ and COG2R: 5′–TCG ACG ACA TCA TTC ACA–3′ were used as forward (COG2F) and reverse (COG2R) primer sequences, respectively [17]. The probe (RING2) was labeled with the fluorescence 5′–FAM and the quencher fluorophore 3′–TAMRA and was 5′–TGG GAG GGC GAT CGC AAT CT–3′. The positive control was HuNoV GII. 4 RNA, while the negative control was RNase-free water [9].

### 2.7. Measurement of pH, Sensory Quality, and Hunter Color of Raw Crabs

After the HPP treatment, pH measurements were taken by diluting 3 g of raw crab in a sterile glass beaker with 27 mL water and stirring for 5 min at room temperature [1,17]. A pH meter (Orion Star A211, Thermo Scientific, Ann Arbor, MI, USA) was used to measure the pH level. Using the hedonic scale, the samples were assessed for color, flavor, taste, texture, and overall acceptability. The following seven-point hedonic scale was used to evaluate the quality: “1”, extreme dislike, unacceptable; “4”, neither liked nor disliked, lower limit of the permissible range; and “7”, extreme liking, virtually defect-free, original quality intact [1]. A score of four or more suggested that the food product was more acceptable. Under identical conditions, the panelists independently evaluated the samples.

To check the color changes in the raw crab following HPP treatment, the Hunter color was assessed [17]. Using a color difference meter, the results revealed the original Hunter color of the standard plates (“L” = 98.48, “a” = 0.14, and “b” = 0.41), as well as the Hunter color difference of the samples as “L (lightness; 0 = dark, 100 = bright)”, “a (redness+, greenness–)”, and “b (yellowness+, blueness–) (UltraScan PRO, Hunterlab, VA, USA) [16]. Because the biological features of raw crab cause the colors of items to differ, all the raw crab samples used in the test were combined. The test was performed three times, with the average value documented for each mixed sample (3 g) [17].

### 2.8. Statistical Analysis

For all studies, each experiment was replicated three times. All data were expressed as mean ± the standard deviation (SD). In the software SPSS version 12.0 (SPSS Inc., Chicago, IL, USA), we used a one-way ANOVA and Duncan’s multi-range test, and the significance of the difference was validated at the possibility level of 5% (*p* < 0.05).

## 3. Results

### 3.1. Effect of HPP on HuNoV GII.4 in Raw Crab Using Non-PMA/RT-qPCR and PMA + Sarkosyl/RT-qPCR

After treating the samples with PMA/sarkosyl, RT-qPCR was used to detect infectious HuNoV. The PMA/sarkosyl treatment led to a greater decrease in the reduction in HuNoV levels compared with the control group. HPP (200, 400, and 600 MPa) was used for 5 min to investigate treatment-related decreases in HuNoV GII.4 in raw crab. In both non-PMA-treated (a–d) and PMA + sarkosyl-treated (e–g) samples, the inoculation of HuNoV GII.4 in raw crab was significantly reduced (*p* < 0.05). Furthermore, when treated with HPP, HuNoV titers were considerably lower in non-PMA-treated samples than in PMA + sarkosyl-treated samples (*p* < 0.05). (Table 1 and Figure 1). Treatment with HPP at 200, 400, and 600 MPa reduction titers were in non-PMA- (0.15, 1.02, and 1.91 log) and PMA + sarkosyl-treated (0.64, 1.8, and 2.34 log), respectively.

### 3.2. Changes in the pH, Sensory Analysis, and Hunter Color with HPP Treatment

The quality parameters (pH, sensory, and Hunter color value) of raw crab were assessed to further examine the applicability of this HPP-treatment decontamination strategy. The pH values (7.48–7.28) among the treated samples differed significantly (*p* < 0.05). (Table 2). The color, smell, taste, and overall acceptability of raw crab were evaluated by untrained panelists to determine the temperature effects on sensory attributes (Table 3). There were not found to be significant differences in sensory evaluations across the treated samples (*p* > 0.05). A study was conducted to investigate the impact of HPP on the Hunter color values (‘L’, ‘a’, and ‘b’) of raw crab subjected to pressure treatment. Changes in color readouts were examined as a function of HPP (200, 400, and 600 MPa for 5 min) (Table 4). Hunter color ‘b’ values reduced significantly (*p* < 0.05) with a stepwise increase in the HPP. Hunter color ‘L’- and ‘a’ values increased significantly (*p* < 0.05) with stepwise increases in HPP experimental pressure.

## 4. Discussion

This study used a real-time reverse transcription polymerase chain reaction (RT-qPCR). The detection of both viable and nonviable viruses is a specific problem for RT-PCR, but, in fact, it affects traditional PCR as well [17]. PMA forms a covalent bond with DNA and RNA in damaged cells or virus particles [17]. Using these qualities, PMA enters the nonviable virus and joins with DNA to discriminate between viable and nonviable viruses. Sarkosyl, an anionic surfactant that aids PMA penetration into injured cells, is another way [19]. Sarkosyl promotes the penetration of PMA through the outer membrane of cells, according to investigations on the NoV and hepatitis viruses [13]. Furthermore, it reports [17,20,21] on HuNoV detection using PMA/sarkosyl and RT-qPCR found that PMA/sarkosyl treatment is a superior way to differentiate infectious viruses by boosting the penetration impact. In this investigation, PMA/sarkosyl were combined to identify only infectious viruses.

To mitigate the risk of HuNoV contamination in seafood, it is crucial to follow strict food safety protocols. This includes ensuring that seafood is sourced from approved and regulated suppliers who adhere to proper harvesting and handling practices. Proper cooking techniques, such as thorough cooking of seafood to appropriate internal temperatures, can effectively reduce the risk of HuNoV transmission. Implementing good hygiene practices throughout the entire seafood supply chain, including regular handwashing, proper sanitation of equipment and surfaces, and using clean water for processing, is vital in preventing HuNoV contamination [21]. The survival of an HuNoV by HPP was examined in depth in this study. HuNoV was reported to be successfully inactivated under optimal HPP conditions with minimal impact on raw crab product quality. We also gave new mechanistic insights into how HPP inactivates viruses. Our findings show that HPP could be a unique intervention tool for raw crab processing. Consumer perceptions of food safety, on the other hand, are influenced by food quality as well as microbial safety. HPP can efficiently inactivate HuNoV, according to our findings. Refrigeration temperature, neutral pH, 200–600 MPa treatment pressure, and a 5 min holding duration are the best conditions for inactivating HuNoV. In raw crab, virus reductions of 0.15 to 1.91 log were achieved without affecting the physical quality of the food samples under these conditions. It should be emphasized that the different viruses’ susceptibilities to HPP differ greatly. FCV, for example, may be effectively inactivated (5 log decreases) at 300 MPa for 3 min at room temperature [22,23]. It has been reported that at 300 MPa and 25 °C for 2 min in cell culture media, an 8 log viral decrease was achieved for rotavirus [24,25]. HAV can reduce cell culture medium by more than 6 log at 450 MPa for 5 min at room temperature [26,27]. Surprisingly, at 600 MPa and ambient temperature for 5 min in cell culture media, no virus decrease was detected for poliovirus or Aichi virus [28]. These findings imply that HPP is promising for inactivating most prevalent agents with minimum influence on food quality. At 4 °C in aqueous medium, an 8.1 log viral decrease was reported under 350 MPa and pH 7.0. However, at 20 °C at the same pressure and pH, only a 4.1 log viral reduction was achieved. All fresh produce showed roughly 5–7 log viral reductions under 450 MPa at 4 °C, but only 4–5 log reductions at 20 °C [25].

While extensive research has shed light on the molecular understanding of how HPP inactivates bacterial pathogens, its specific mechanism for virus inactivation remains unknown. It is important to note that HPP has limited impact on covalent bonding, which raises questions about its ability to break down viral genomic RNA [25,29,30,31]. Studies have suggested that HPP may cause viral protein damage, but the detailed experimental evidence to establish the exact mechanism is lacking [22,26,27,29]. Further experimental investigations are needed to unravel the precise mechanisms through which HPP inactivates viruses and understand its potential effects on viral genetic material. Tang et al. ([32]) previously demonstrated that HPP-treated MNV-1 had a large concentration of the VP1 gene. This is not surprising, given the fact that they reported a 3 log virus survival rate after HPP treatment [25]. Sánchez et al. (2011) employed a quantitative RT-qPCR approach to compare MNV and HuNoV genogroup II.4 survival and found that HuNoV RNA was more HPP-resistant (up to 500 MPa) than MNV RNA [33]. However, it is unclear whether their tests were conducted in an RNase-free environment. When analyzing RNA samples from completely inactivated viruses following RT-PCR at 600 MPa for 2 min, both untreated and treated viruses exhibited identical amplification levels of the VP1 gene [25]. When viral genomic RNA was directly treated with HPP, a similar outcome was found. These findings showed that HPP did not destroy viral genomic RNA at pressures ranging from 350 to 600 MPa for 2 min [25]. These findings show that damage to the viral capsid structure, rather than genomic RNA, is the major mechanism of MNV-1 inactivation by HPP [25]. HPP at 350 MPa had no effect on the sensory aspects of the fresh produce examined, such as look, color, or aroma, except lettuce; at 600 MPa, lettuce, raspberries, and strawberries showed a more significant textural loss. Many NoV outbreaks have been linked to frozen fruits, such as blueberries, raspberries, strawberries, and grapes, over the last two decades [30,34]. Salads, cakes, purees, juices, smoothies, creams, ice creams, and other goods using these fruits are all popular. Viruses also have a longer shelf life when they are frozen or cooled [35,36]. Reduced temperature and decreased water activity, which are employed to regulate bacterial populations, have no effect on viruses [34,37]. Even after 6 months of freezing, no decrease in MNV-1 was found in frozen onions and spinach [34]. The color, taste, texture, appearance, and smell of seafood (raw oyster) was much better when treated with HPP [38]. HPP’s ability to effectively inactivate both surface and internalized viruses make it ideal for usage in the seafood industry.

Therefore, our research found that HPP successfully inactivated HuNoV under ideal processing conditions of moderate pressure, pH, and sensory attributes of raw crabs. HPP could be an auspicious application for eradicating and reducing HuNoV risk in raw crab and its associated products while causing minimal food-quality degradation.

## 5. Conclusions

In conclusion, our findings show that under ideal processing circumstances, HPP effectively inactivates HuNoV. In this study, the inactivation of HuNoV in raw crabs was investigated using HPP at 200, 400, and 600 MPa for 5 min. The raw crab subjected to 200–600 MPa of HPP showed considerable changes in Hunter color and pH, as well as sensory characteristics. Because of our findings, 600 MPa of HPP could be an effective treatment for raw crabs to improve the physical color and pH of the meat. HPP could be a probable application for eradicating and reducing HuNoV risk in raw crabs and related goods with a low food-quality impact. HPP can indeed effectively reduce HuNoV in crab meat contamination. However, if further measures are required to ensure complete safety, additional steps can be taken. These steps might involve implementing strict hygiene practices during processing, and for surfaces and utensils. Furthermore, proper handling and storage conditions, including maintaining appropriate temperature controls, can contribute to reducing the risk of HuNoV contamination. It is essential to follow industry guidelines and regulations to guarantee the highest level of food safety [39].

## Figures and Tables

**Figure 1 viruses-15-01599-f001:**
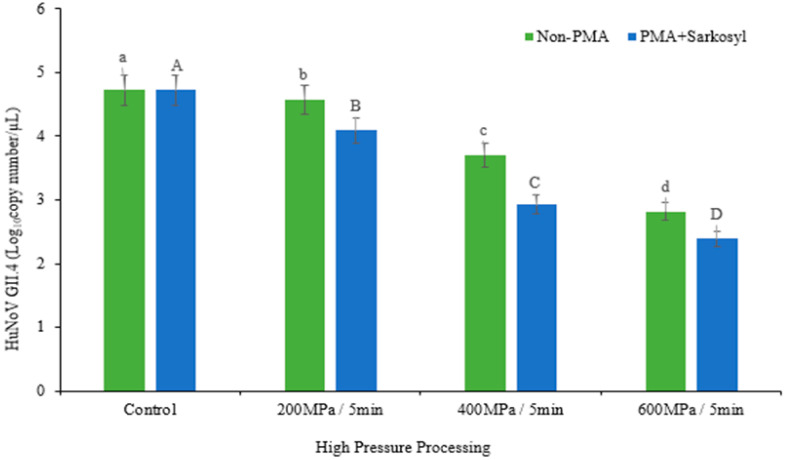
Reduction in HuNoV GII.4 titers after inactivation with high-pressure processing at 200, 400, 600 MPa for 5 min without PMA and with PMA + sarkosyl. According to Duncan’s multiple range test with different letters (a–d) and (A–D), all data were significantly different (*p* < 0.05). The number of replicates (*n* = 3).

**Table 1 viruses-15-01599-t001:** Effects of HPP treatment against HuNoV GII.4 d with/without propidium monoazide (PMA) or sarkosyl of raw crabs.

Treatment	Non-PMA	PMA + Sarkosyl
	log_10_copy Number/µL	log_10_copy Number/µL
Control	4.73 ± 0.06 ^a^	4.73 ± 0.06 ^a^
200 MPa	4.58 ± 0.08 ^b^	4.09 ± 0.01 ^e^
400 MPa	3.71 ± 0.05 ^c^	2.93 ± 0.00 ^f^
600 MPa	2.82 ± 0.03 ^d^	2.39 ± 0.00 ^g^

Within the same column and row, HuNoV log reduction means with SDs. According to Duncan’s multiple range test with different letters (a–g), all data were significantly different (*p* < 0.05). The number of replicates (*n* = 3).

**Table 2 viruses-15-01599-t002:** Effects of high-pressure processing on the pH value of raw crabs.

Treatment	High-Pressure Processing			
	Control	200 MPa	400 Mpa	600 Mpa
pH	7.48 ± 0.02 ^a^	7.42 ± 0.05 ^ab^	7.38 ± 0.03 ^b^	7.28 ± 0.02 ^c^

Within the same row, the data indicate means with SDs (three samples/treatment) with different letters (a–c). Data were significantly different (*p* < 0.05) according to Duncan’s multiple range test. The number of replicates (*n* = 3).

**Table 3 viruses-15-01599-t003:** Effects of high-pressure processing on HuNoV GII.4 in raw crabs by sensory evaluation (color, smell, taste, appearance, and overall acceptability).

Treatment	Sensory Evaluation				
	Color	Smell	Taste	Appearance	Overall Acceptability
Control	5.30 ± 0.60	5.30 ± 1.10	5.30 ± 0.80	5.60 ± 0.60	5.50 ± 1.50
200 Mpa	5.20 ± 0.80	5.40 ± 0.60	5.20 ± 0.60	5.50 ± 0.80	5.20 ± 1.00
400 Mpa	5.00 ± 0.70	5.40 ± 0.90	5.20 ± 0.60	5.50 ± 0.90	5.10 ± 0.70
600 Mpa	4.70 ± 0.60	5.10 ± 0.50	5.10 ± 1.60	5.20 ± 1.05	4.90 ± 1.40

Within the same column, the data indicate means with SDs (three samples/treatment). According to Duncan’s multiple range test, all data were not significantly different (*p* > 0.05). The number of replicates (*n* = 3).

**Table 4 viruses-15-01599-t004:** Effects of high-pressure processing on HuNoV GII.4 in raw crabs by Hunter colors.

Treatment	Hunter Colors		
	‘L’ Value	‘a’ Value	‘b’ Value
Control	22.30 ± 0.22 ^d^	8.85 ± 0.02 ^d^	13.92 ± 0.06 ^a^
200 Mpa	28.91 ± 0.03 ^c^	12.69 ± 0.03 ^c^	10.89 ± 0.06 ^b^
400 Mpa	40.54 ± 0.12 ^b^	15.49 ± 0.05 ^b^	8.81 ± 0.10 ^c^
600 Mpa	41.85 ± 0.05 ^a^	16.67 ± 0.03 ^a^	7.78 ± 0.04 ^d^

Within the same column, the data indicate means with SDs (three samples/treatment) with different letters (a–d). According to Duncan’s multiple range test, all data were significantly different (*p* < 0.05). Hunter “L” values = whiteness+, darkness–; Hunter “a” values = redness+, greenness–; Hunter “b” values = yellowness+, blueness–. The number of replicates (*n* = 3).

## Data Availability

Data are contained within the article. The data presented in this study are available on request from the corresponding author.

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
