# Peer review of "Application of High-Pressure Processing (or High Hydrostatic Pressure) for the Inactivation of Human Norovirus in Korean Traditionally Preserved Raw Crab"

_viruses, 2023, doi:10.3390/v15071599_

Round 1

Reviewer 1 Report

Title: Application of high-pressure processing (or high hydrostatic pressure) for the inactivation of human norovirus in the Korean traditional preserved raw crab

Authors: Kumar Roy et al.

Summary

This manuscript aims to find a methodology to sterilize HNoV from raw crab meat used in traditional Korean cuisine. For this purpose, the authors used a high-pressure processing technique widely used in the food industry to sterilize food from diverse pathogens, including bacteria and viruses. After HPP treatment of HNoV-contaminated raw crab meat, the virus genome was quantified using qRT-PCR of samples pre-treated without or with PMA/ sarkosyl to distinguish between no intact and intact virus particles (infectious). The HPP treatment does not affect the food properties of the raw crab meat, such as pH, color, smell, taste, appearance, and acceptability, even though some variations are observed with the Hunter colors detection method.

Comments

1)    The main point of this manuscript corresponds to the RT-qPCR performed on crab meat after being treated at diverse HPPs (200, 400, and 600 MPa). To detect HNoV viruses, the samples were untreated or treated with PMA/sarkosyl to determine all RNA or RNA from intact particles, respectively. Since HPP is described to denature capsid proteins, it is expected then to have fewer intact particles at increasing HPP conditions compared to the control samples (no HPP). However, what intrigues me, is that at increasing HPP, the total amount of virus RNA corresponding to intact particles and disrupted particles (no PMA treatment) should be the same between control and HPP-treated samples. In the results presented in this manuscript, this value are decreasing concomitantly with the increase in HPP treatment. Please, explain this observation. Is the virus RNA genome damaged, too, with HPP treatment? If it is so, please provide evidence.

2)    Pg 2, 1st paragraph: Describe in the introduction what you mean by non-infectious and infectious virus. You need to clarify since the beginning that you are identifying intact and no-intact virus particles.

3)    Pg 2 and whole manuscript: Many times in the text, we can read: …As a result, the result…. This is redundant; please change it

4)    In the third paragraph, remove: “…after they were treated with HPP…since redundant”.

5)    Pg 3, 1st paragraph: I do not see the point of the description of the sauces (even if culinarily interesting) used to prepare the traditional Korean crab. They are not relevant to the topic of research. Please, remove it from the paragraph.

6)    Pg 3, 3rd paragraph: change “sequestered“to “obtained.”

7)    Pg 3, 4th paragraph: change “freezer (4ºC)” to “fridge (4ºC)”.

8)    Pg 3, 4th paragraph: change “ for about an h” to “for 1 h”

9)    Pg 3, 6th paragraph: Please specify the distance between the sample and the LED light used after PMA treatment.

10) Pg 4, 2nd paragraph. Please provide the name of the enzyme used for RT-PCR and the brand of the kit used.

11) Pg 4, 3rd paragraph: Add reference after mentioning the hedonic scale.

12) Pg 5, 1st paragraph:  change to “p< 0.05”.

13) Pg 5, 2nd paragraph HNV change to HNoV to be consistent with the rest of the manuscript.

14) Pg 5, 2nd paragraph (last sentence). This sentence makes no sense with your results because you are not checking the effect of the sarkosyl addition when applied HPP. For this purpose, you must compare your sample upon HPP with no-PMA, PMA, and PMA-sarkosyl and then determine the virus titer.

15) Pg 6, Please, in the legend of table 3, indicate the number of subjects that performed the sensory evaluation. It is unclear from the text.

16) In general, the discussion is very long and redundant. It should go directly into discussing your results and perhaps compare them with a few other similar cases. But, currently is more confusing than clarifying the results obtained from this research.

17) Please indicate if the treatment described in this manuscript can be effectively used to sterilize meat crab from HNoV or if it requires more work, which will be the following step.

Please check the comments above.

Reviewer 2 Report

The manuscript entitled “Application of high pressure processing (or high hydrostaic pressure) for the inactivation of human norovirus in the Korean traditional preserved raw crab” shows the results of the use on the inactivation of NoV by HPP method in seafood. The manuscript is generally well written however there are some comments and mistakes that should be addressed before publication.

See attached file

Round 2

Reviewer 2 Report

The revised article “Application of high pressure processing (or high hydrostaic pressure) for the inactivation of human norovirus in the Korean traditional preserved raw crab” is more understandable and clearer than the previous version. However, there are still some issues in the manuscript that need to be modified.

Abstract

Line 22: You wrote…..crabs were significantly different (p>0.05) ….

But if it is significan p should be <0.05. please correct it.

Introduction

Lines 54: this sentence speaks of vegetables has nothing to do with the previous sentences. Please delete it.

Lines 56: in the same way, here we are talking about fresh produce, therefore fruit and vegetables. Please here too you should talk about seafood.

Line 72: …..living….

This term does not seem appropriate for viruses. Please change it with viable.

Results

Table 1 line 224 and Figure 1 line 230: in the caption below the table and figure you wrote…..all data were significantly different (p>0.05) ….

But if it is significan p should be <0.05. please correct it.

Discussion

Lines 260 and 263: Living and dead virus is not suitable. Please change the terms with “viable and nonviable” or with “infectious or non-infectious”.

Line 262: dead virus…

Also this term does not seem appropriate for viruses. Please change it.

Conclusion

Line 344 and 345: you use the terms sterilize and eliminate HuNoV. Sterilization destroys all microorganisms present on an object or matrix, but HPP it doesn't do this. Moreover, HuNoV in your study is not removed completely. The HPP treatment you described reduced the virus not eliminated so these two terms you used are incorrect. Please change them.
